# A Bridge from Audio to Video: Phoneme-Viseme Alignment Allows Every Face to Speak Multiple Languages

## Abstract

Speech-driven talking face synthesis (TFS) focuses on generating lifelike facial animations from audio input. Current TFS models perform well in English but unsatisfactorily in non-English languages, producing wrong mouth shapes and rigid facial expressions. The terrible performance is caused by the English-dominated training datasets and the lack of cross-language generalization abilities. Thus, we propose Multilingual Experts (**MuEx**), a novel framework featuring a Phoneme-Guided Mixture-of-Experts (**PG-MoE**) architecture that employs phonemes and visemes as universal intermediaries to bridge audio and video modalities, achieving lifelike multilingual TFS. To alleviate the influence of linguistic differences and dataset bias, we extract audio and video features as phonemes and visemes respectively, which are the basic units of speech sounds and mouth movements. To address audiovisual synchronization issues, we introduce the Phoneme-Viseme Alignment Mechanism (**PV-Align**), which establishes robust cross-modal correspondences between phonemes and visemes. In addition, we build a Multilingual Talking Face Benchmark (**MTFB**) comprising 12 diverse languages with 95.04 hours of high-quality videos for training and evaluating multilingual TFS performance. Extensive experiments demonstrate that MuEx achieves superior performance across all languages in MTFB and exhibits effective zero-shot generalization to unseen languages without additional training.

## 1 Introduction

Speech-driven TFS has made remarkable progress in creating realistic facial animations with precise lip synchronization Prajwal et al. (2020); Zhang et al. (2023); Cui et al. (2025); Wang et al. (2025). These advances enable applications ranging from virtual assistants to content creation. However, current methods face severe limitations when processing non-English speech.

As shown in Figure 1, three critical issues emerge when existing TFS models Zhang et al. (2023); Wei et al. (2024); Cui et al. (2025) process non-English languages. First, there are mismatches between phonemes (basic units of speech sounds) and visemes (corresponding mouth shapes) when models process new audio patterns, as shown in Figure 1 (a). Second, audiovisual decoupling leads to a more severe issue: clear speech audio fails to produce corresponding mouth movements, as shown in Figure 1 (b). Third, facial expressions become rigid, especially in tonal languages (Chinese), as shown in Figure 1 (c).

The root cause of the above audiovisual mismatches stems from two critical flaws in existing methods. First, existing models are mainly trained on English datasets like VoxCeleb Nagrani et al. (2017), CelebV-HQ Zhu et al. (2022), and HDTF Zhang et al. (2021), resulting in generation patterns optimized specifically for English audiovisual alignment. When processing non-English speech with different phonetic structures, the pre-trained audiovisual alignment fails to generate accurate mouth movements from multilingual speech. Second, existing approaches construct audiovisual alignment through end-to-end learning without modeling the underlying phonological principles. Thus, these methods lack the ability to generalize to multilingual audio, resulting in incorrect mouth movements and poor audiovisual synchronization.

Therefore, we propose **MuEx**, a novel framework featuring a phoneme-guided mixture-of-experts **(PG-MoE)** architecture that employs phonemes and visemes as universal intermediaries to bridge audio and video modalities, achieving lifelike multilingual TFS. Our approach centers on two core innovations. First, we introduce the **PV-Align** mechanism that creates language-agnostic representations by extracting phonemes from audio and visemes from video, then establishing robust cross-modal correspondences through adversarial learning and clustering. This resolves phoneme-viseme mismatches by learning universal audiovisual correspondences that transcend linguistic boundaries. Second, we design a **PG-MoE** that leverages these universal phoneme-viseme alignments for MoE routing. The system selects experts based on audiovisual similarities rather than language labels, enabling the model to choose appropriate processing pathways for multiple languages without explicit supervision.

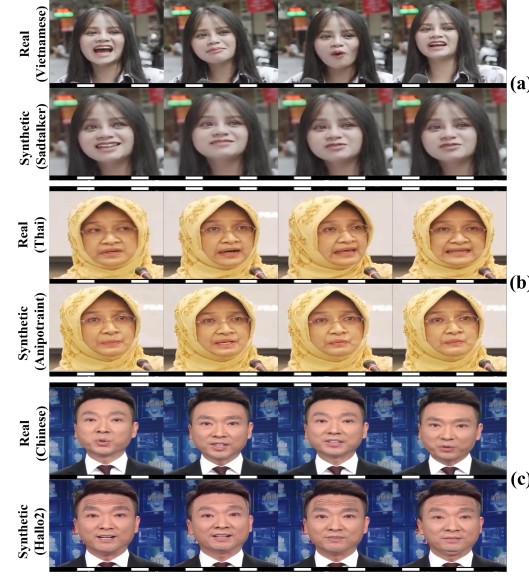

Figure 1: Comparison of real and TFS videos on non-English languages reveals phoneme-viseme mismatches, rigid facial expressions, and audio-visual decoupling.

Extensive experiments demonstrate that MuEx achieves superior audiovisual synchronization across all languages in MTFB. Furthermore, MuEx shows effective zero-shot generalization to unseen languages without extra training. Our framework can generate lifelike talking face videos from any language audio without requiring language labels.

**Our contributions include:**

- We design a novel speech-driven face synthesis paradigm **(MuEx)** utilizing phonemes and visemes as intermediaries to bridge audiovisual modalities, overcoming multilingual TFS generalization barriers and cross-modal synchronization challenges.

- We propose the Phoneme-Viseme Alignment **(PV-Align)** mechanism to establish robust cross-modal correspondences between phonemes and visemes, thereby enabling effective zero-shot generalization to unseen languages.

- We introduce a comprehensive Multilingual Talking Face Benchmark **(MTFB)** comprising 12 languages with 95.04 hours of high-quality audiovisual content for training and evaluating multilingual TFS performance.

## 2 RELATED WORK

### 2.1 AUDIO-DRIVEN TALKING HEAD GENERATION

Recent advances in audio-driven talking head generation have leveraged deep learning for improved realism. GAN-based methods like Wav2Lip Prajwal et al. (2020) and SadTalker Zhang et al. (2023) achieved strong lip-sync accuracy and facial expression control. Diffusion-based approaches such as DiffTalk Shen et al. (2023) and Hallo2 Cui et al. (2025) demonstrated high-quality facial animation synthesis. Transformer architectures including AniTalker Peng et al. (2024) and EchoMimic Chen et al. (2024) focused on temporal consistency and identity preservation. However, these methods primarily target single-language scenarios and struggle with multilingual generalization.

### 2.2 MULTILINGUAL SPEECH AND MULTILINGUAL LEARNING

Multilingual speech processing has made progress with self-supervised methods like Wav2Vec 2.0 Baevski et al. (2020) and XLSR Conneau et al. (2020), demonstrating multilingual transfer capa-

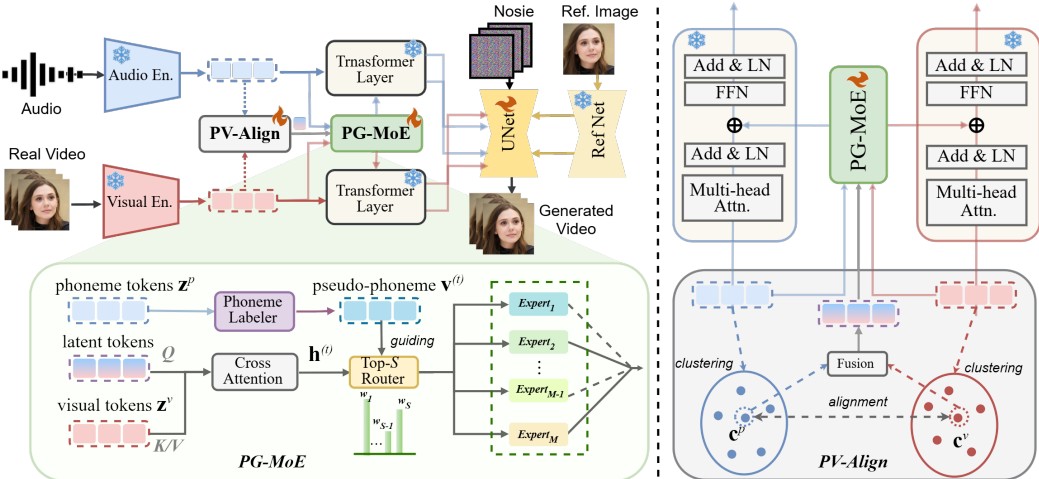

Figure 2: Framework. The model learns universal viseme–phoneme prototypes and employs pseudo-phoneme guided expert routing to enable cross-lingual speech-driven TFS.

bilities across diverse languages. Understanding phoneme-to-viseme correspondences is crucial for lip-sync generation, with research showing that languages exhibit distinct articulatory patterns and viseme distributions Massaro & Cohen (1999). multilingual transfer learning techniques, including few-shot learning Finn et al. (2017) and zero-shot generalization Radford et al. (2021), have shown promise for handling low-resource scenarios, while continual learning methods address catastrophic forgetting Kirkpatrick et al. (2017).

### 2.3 MIXTURE OF EXPERTS AND DISCRETE REPRESENTATION

Mixture of Experts (MoE) architectures have proven effective for handling diverse tasks through sparse expert routing, as demonstrated in Switch Transformer Fedus et al. (2022) and Vision MoE Riquelme et al. (2021). For facial animation, discrete representation learning through vector quantization has gained attention. VQTalker Liu et al. (2025) introduced facial motion quantization for talking head generation, while methods like VQ-VAE Van Den Oord et al. (2017) established foundations for discrete generative modeling.

Despite advances in TFS, existing methods struggle with multilingual scenarios, exhibiting phoneme-viseme mismatches and audiovisual decoupling when processing non-English speech. These English-centric approaches lack cross-language generalization and require complete retraining for different languages. Our MuEx framework addresses these limitations through PG-MoE and Pv-Align, enabling robust multilingual synthesis with zero-shot transfer capabilities.

## 3 METHOD

### 3.1 OVERVIEW

Our goal is to improve cross-lingual generalization in speech-driven TFS. A key challenge lies in the variability of speech-to-lip correspondences across different languages, which often prevents models trained on one language from transferring effectively to others. To overcome this, we propose a framework that disentangles and aligns phoneme and viseme representations in a language-agnostic manner. As shown in Figure 2, the pipeline consists of three major components. First, we cluster speech (phoneme-level) and visual (viseme-level) features into prototypes, which act as universal anchors capturing articulatory units shared across languages. Second, we align phoneme and viseme prototypes by maximizing their mutual information while suppressing spurious correlations at the raw feature level, ensuring that the learned mapping reflects stable and transferable articulatory patterns. Third, we estimate pseudo-phonemes and use them to guide a sparse mixture-of-experts (MoE) router, which dynamically selects specialized expert modules to synthesize natural

and synchronized lip movements in multiple languages. Together, these components form a unified architecture that builds a universal viseme–phoneme mapping and leverages it for cross-lingual speech-driven TFS.

## 3.2 PROTOTYPE-BASED PHONEME–VISEME ALIGNMENT

A central step in our framework is to construct language-independent articulatory anchors that bridge the gap between speech and visual modalities. To this end, we cluster speech features at the phoneme level and visual features at the viseme level into $K$ prototypes, denoted as $\{\mathbf{c}_k^p\}_{k=1}^K$ for phonemes and $\{\mathbf{c}_k^v\}_{k=1}^K$ for visemes. We set $K$ close to the number of phonemes ($K = 40$) and initialize prototypes using K-means++ to achieve stable coverage of the feature space. Prototypes are periodically updated (every 10 epochs in our implementation) to adapt to the evolving representation space without incurring high computational cost.

Given an input feature $\mathbf{z}_t^p$ or $\mathbf{z}_t^v$ at time step $t$, we assign it to the nearest prototype to obtain discrete codes $\mathbf{q}_t^p$ and $\mathbf{q}_t^v$. This hard assignment makes the mapping interpretable, while a parallel soft assignment $\mathbf{v}^{(t)} = \mathrm{softmax}(-\|\mathbf{z}_t^p - \mathbf{c}_k^p\|^2/\tau^2)$ with temperature $\tau$ provides smoother gradients. Intuitively, prototypes act as universal anchors: although phoneme inventories vary across languages, articulatory gestures such as lip closure, tongue elevation, or jaw movement recur across languages, allowing prototypes to capture these transferable units.

As shown in Figure. 3, we enforce a meaningful alignment between phoneme and viseme spaces with a mutual-information (MI) objective:

$$\mathcal{L}_{\mathrm{align}} = - I^{\mathrm{JS}}\big(\mathbf{q}^p, \mathbf{q}^v\big) \ + \ \lambda_{\mathrm{neg}} \, I^{\mathrm{JS}}\big(\mathbf{z}^p, \mathbf{z}^v\big), \tag{1}$$

where the first term maximizes the Jensen–Shannon MI between matched phoneme–viseme *prototypes* $(\mathbf{q}^p, \mathbf{q}^v)$, and the second term discourages spurious correlations at the *raw-feature* level $(\mathbf{z}^p, \mathbf{z}^v)$.

To estimate MI consistently across both terms, we adopt the Jensen–Shannon lower bound with a binary-classifier view. Let $\mathcal{P}$ denote the set of matched pairs and $\mathcal{N}$ the set of shuffled (mismatched) pairs. Using a lightweight discriminator $D_\psi$ that takes the concatenation $[\mathbf{x}; \mathbf{y}]$ as input, the estimator is

$$I^{\mathrm{JS}}(\mathbf{x}, \mathbf{y}) = \mathbb{E}_{(\mathbf{x},\mathbf{y})\sim\mathcal{P}}\big[\log \sigma\big(D_\psi([\mathbf{x}; \mathbf{y}])\big)\big] + \mathbb{E}_{(\mathbf{x},\mathbf{y})\sim\mathcal{N}}\big[\log\big(1 - \sigma\big(D_\psi([\mathbf{x}; \mathbf{y}])\big)\big)\big], \tag{2}$$

where $\sigma(\cdot)$ denotes the logistic sigmoid.

This design yields two advantages. First, discrete prototypes provide stable cross-lingual anchors that remain consistent across languages. Second, the MI-based objective ensures that alignment occurs at the prototype level rather than raw features, promoting language-agnostic correspondences. Together, these properties enable our talking face generator to synthesize realistic and synchronized lip movements even when applied to languages unseen during training.

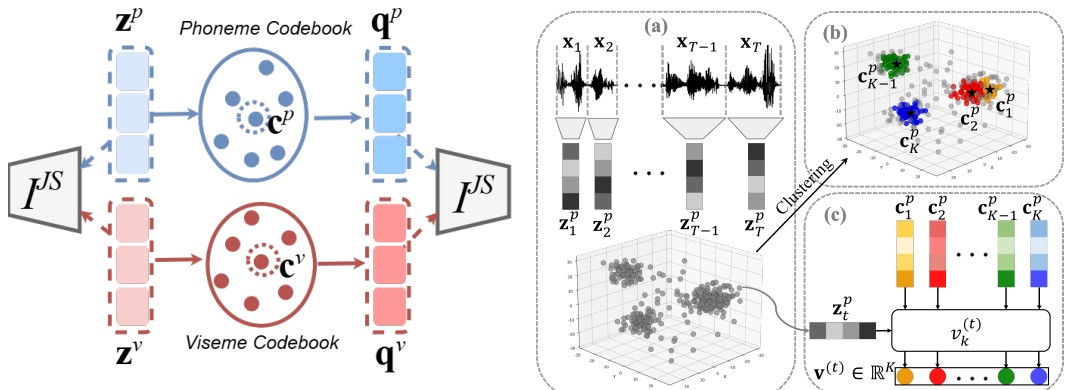

Figure 3: Illustration of phoneme-viseme aligning via Jensen–Shannon MI.

Figure 4: Mechanism of the Phoneme Labeler.

## 3.3 PSEUDO-PHONEME GUIDED EXPERT ROUTING

While prototype alignment provides universal anchors, it is still necessary to dynamically adapt feature processing across languages. To this end, we introduce a sparse mixture-of-experts (MoE) router guided by pseudo-phoneme labelss. At each time step $t$, the router combines content features $\mathbf{h}^{(t)}$ with the pseudo-phoneme labels $\mathbf{v}^{(t)}$ to produce expert scores $s_i = \beta\, g_{\mathrm{phon},i}(\mathbf{v}^{(t)}) + (1 - \beta)\, g_{\mathrm{cont},i}(\mathbf{h}^{(t)})$, where $\beta \in [0, 1]$ balances phoneme- and content-based gating. The top-$S$ experts are selected, and their outputs are aggregated with softmax-normalized weights.

The mechanism of the Phoneme Labeler is shown in Figure 4, we define the soft assignment probability for each phoneme cluster based on the similarity between the audio-visual embedding $\mathbf{z}_t^p$ and phoneme centroids $\mathbf{c}_k^p$, using a temperature-scaled softmax function. The pseudo-phoneme labels $\mathbf{v}^{(t)}$ is then given by:

$$\mathbf{v}^{(t)} = \mathrm{softmax}\left(\mathcal{A}(\mathbf{z}_t^p, \mathbf{c}_k^p)\right), \tag{3}$$

where $\mathcal{A}(\mathbf{z}_t^p, \mathbf{c}_k^p) = -\frac{\|\mathbf{z}_t^p - \mathbf{c}_k^p\|_2^2}{\tau^2}$ defines the phonetic affinity between the embedding $\mathbf{z}_t^p$ and the phoneme centroid $\mathbf{c}_k^p$, and $\tau$ is a temperature parameter that controls the concentration of the affinity. Thus, the full pseudo-phoneme labels is $\mathbf{v}^{(t)} = (v_1, v_2, \ldots, v_K)^\top$.

To encourage consistent and balanced expert usage, we impose a routing loss:

$$\mathcal{L}_{\mathrm{router}} = \mathcal{L}_{\mathrm{route}} + \lambda_{\mathrm{util}} \frac{M}{B} \sum_{i=1}^{M} \left(\frac{n_i}{B}\right)^2 - \lambda_{\mathrm{ent}} \sum_{i \in \mathcal{Q}} w_i \log w_i, \tag{4}$$

where $\mathcal{L}_{\mathrm{route}}$ is a masked cross-entropy between the selected experts and the pseudo-phoneme labels, $n_i$ is the usage count of expert $i$ in a batch of size $B$, and $\mathcal{Q}$ denotes the selected experts.

By leveraging pseudo-phoneme supervision, the router learns to activate experts aligned with cross-lingual articulatory patterns, allowing the generator to synthesize more accurate and natural lip movements in multiple languages.

## 3.4 TRAINING OBJECTIVE AND WORKFLOW

Our model jointly optimizes prototype alignment, expert routing, and the TFS objective. The overall loss is defined as

$$\mathcal{L} = \mathcal{L}_{\mathrm{align}} + \mathcal{L}_{\mathrm{router}} + \lambda_{\mathrm{task}} \mathcal{L}_{\mathrm{gen}}, \tag{5}$$

where $\mathcal{L}_{\mathrm{align}}$ enforces universal phoneme–viseme mapping, $\mathcal{L}_{\mathrm{router}}$ guides expert activation with pseudo-phoneme supervision, and $\mathcal{L}_{\mathrm{gen}}$ drives the talking face generator.

**Generation loss.** To ensure both pixel fidelity and perceptual realism, we define

$$\mathcal{L}_{\mathrm{gen}} = \lambda_1 \|\hat{\mathbf{I}} - \mathbf{I}\|_1 + \lambda_p \|\phi(\hat{\mathbf{I}}) - \phi(\mathbf{I})\|_2^2 + \lambda_t \|\nabla_t \hat{\mathbf{I}} - \nabla_t \mathbf{I}\|_1, \tag{6}$$

where $\hat{\mathbf{I}}$ and $\mathbf{I}$ denote the generated and ground-truth video frames, $\phi(\cdot)$ extracts deep features from a pretrained perceptual network (VGG), and $\nabla_t$ denotes temporal frame differences. The first term enforces pixel-level accuracy, the second term preserves semantic and structural realism, and the third term improves temporal smoothness of lip movements. The weights $\lambda_1, \lambda_p, \lambda_t$ control the relative contributions of these objectives.

**Training workflow.** During each training step, we (1) extract phoneme and viseme features from the input speech and video, (2) assign features to prototypes and compute the alignment loss, (3) estimate pseudo-phoneme s and apply MoE routing, (4) synthesize video frames and compute $\mathcal{L}_{\mathrm{gen}}$, and (5) jointly update all modules by minimizing the total loss. This unified optimization enables the model to capture language-agnostic articulatory correspondences while generating realistic cross-lingual talking face videos.

## 4 EXPERIMENTS

### 4.1 DATASET DETAILS

We employ a two-stage training approach using complementary datasets. Initially, we train on the HDTF Zhang et al. (2021) dataset, which provides over 15 hours of high-quality English talking face videos at 512×512 resolution with clear audiovisual synchronization.

For multilingual generalization, we constructed a multilingual talking face benchmark (MTFB) by collecting news broadcasts and interview videos from 12 countries. As shown in figure 5, the dataset contains 95.04 hours of high-quality content spanning diverse language families. Each video segment undergoes careful preprocessing to ensure single-person appearance, frontal face orientation, and clean audio without background music or noise. All videos are standardized to 512×512 resolution to maintain consistency with the HDTF baseline. The collection includes both tonal languages (Chinese, Thai, Vietnamese) and non-tonal languages, providing comprehensive coverage for evaluating multilingual performance. This dataset addresses the lack of diverse multilingual benchmarks in TFS research, particularly for underrepresented languages like Arabic and Thai.

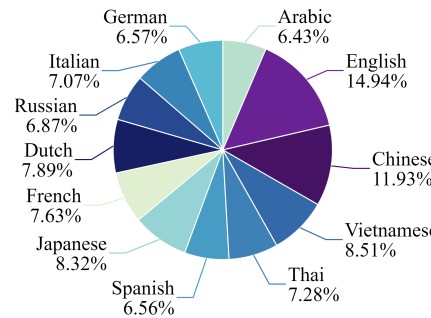

Figure 5: Details of MTFB.

### 4.2 EVALUATION METRICS

We use the Fréchet Video Distance (FVD) Unterthiner et al. (2019) to quantify the distributional differences between generated and real videos. Additionally, we adopt SyncNet Prajwal et al. (2020) and use the SyncNet Confidence (Sync-C) score as a metric to assess audio-lip synchronization quality. To better evaluate multilingual scenarios, we further propose two novel metrics.

#### 4.2.1 LIP-SYNC ERROR DISTANCE (LSE-D)

We propose LSE-D to measure geometric consistency between mouth movements in generated and ground truth videos. We extract 26 key lip landmarks using MediaPipe Face Mesh Kartynnik et al. (2019) with standardized preprocessing for scale invariance. The frame-wise lip-sync error is:

$$\text{LSE}_t = \frac{1}{26} \sum_{i=1}^{26} \|l_{t,i}^{real} - l_{t,i}^{gen}\|_2. \tag{7}$$

The overall LSE-D score is the temporal average: $\text{LSE-D} = \frac{1}{T} \sum_{t=1}^{T} \text{LSE}_t$, where lower values indicate better synchronization.

#### 4.2.2 TEMPORAL MOUTH DYNAMICS CORRELATION (TMDC)

To capture fine-grained temporal consistency across linguistic patterns, we propose TMDC. For each frame $t$, we extract five scale-invariant mouth features from lip landmarks $\mathbf{L}_t$:

$$\mathbf{F}_t = \begin{bmatrix} \|\mathbf{L}_{left} - \mathbf{L}_{right}\|_2 \\ \|\mathbf{L}_{top} - \mathbf{L}_{bottom}\|_2 \\ \text{ConvexHull}(\mathbf{L}_t) \\ \frac{\|\mathbf{L}_{left} - \mathbf{L}_{right}\|_2}{\|\mathbf{L}_{top} - \mathbf{L}_{bottom}\|_2 + \epsilon} \\ \frac{\|\mathbf{L}_{top} - \mathbf{L}_{bottom}\|_2}{\|\mathbf{L}_{left} - \mathbf{L}_{right}\|_2 + \epsilon} \end{bmatrix} = \begin{bmatrix} \text{Width}_t \\ \text{Height}_t \\ \text{Area}_t \\ \text{AspectRatio}_t \\ \text{Openness}_t \end{bmatrix}, \tag{8}$$

where $\epsilon = 10^{-8}$ prevents division by zero.

For each feature $k$, we compute Pearson correlation between real and generated sequences:

$$r_k = \frac{\sum_{t=1}^{T} (f_{k,t}^{real} - \bar{f}_k^{real})(f_{k,t}^{gen} - \bar{f}_k^{gen})}{\sqrt{\sum_{t=1}^{T} (f_{k,t}^{real} - \bar{f}_k^{real})^2} \sqrt{\sum_{t=1}^{T} (f_{k,t}^{gen} - \bar{f}_k^{gen})^2}}. \tag{9}$$

The overall TMDC score is $\text{TMDC} = \frac{1}{5} \sum_{k=1}^{5} r_k$, where higher values indicate better temporal consistency.

## 4.3 EXPERIMENTAL RESULTS

| Language | Methods (Lip-audio Sync/Teeth naturalness Scores) | | | | | | Type |
|---|---|---|---|---|---|---|---|
| | MuEx | EchoMimic | Hallo2 | AniPortrait | SadTalker | AniTalker | |
| English | **5.70/5.60** | 3.15/5.40 | 5.30/1.00 | 2.10/2.20 | 3.75/3.65 | 1.00/3.15 | Seen |
| Chinese | **5.70/5.15** | 4.65/4.75 | 4.55/4.95 | 2.85/2.30 | 1.55/1.60 | 1.70/2.25 | Seen |
| Spanish | **5.55/5.55** | 4.60/4.20 | 4.85/5.20 | 1.75/1.95 | 3.00/2.75 | 1.25/1.35 | Seen |
| French | **5.60**/4.20 | 5.15/4.45 | 4.20/3.85 | 2.85/**5.50** | 1.00/1.05 | 2.20/1.95 | Seen |
| German | **5.65/5.35** | 2.70/2.45 | 4.50/4.80 | 1.35/1.40 | 4.45/4.40 | 2.35/2.60 | Seen |
| Japanese | **5.70/5.70** | 4.80/4.65 | 3.60/1.50 | 1.90/3.55 | 3.55/3.70 | 1.45/1.90 | Seen |
| Arabic | **5.70/5.65** | 3.75/3.80 | 3.65/3.15 | 3.20/3.75 | 2.60/2.40 | 2.10/2.25 | Seen |
| Dutch | **5.65**/5.00 | 4.95/4.65 | 4.25/3.20 | 2.70/**5.10** | 1.05/1.15 | 2.40/1.90 | Seen |
| Russian | **5.80/5.60** | 4.55/3.95 | 4.55/4.55 | 1.90/3.05 | 2.45/2.05 | 1.75/1.80 | Seen |
| Italian | **5.65/5.45** | 3.45/4.55 | 5.15/4.80 | 3.75/3.20 | 1.15/1.70 | 1.85/1.30 | Seen |
| Thai | **5.75/5.85** | 3.50/4.30 | 3.80/3.05 | 4.75/4.60 | 1.10/1.00 | 2.10/2.20 | Seen |
| Vietnamese | **5.65/5.30** | 3.45/5.15 | 5.35/4.55 | 3.55/3.00 | 1.55/1.30 | 1.45/1.70 | Seen |
| Korean | **5.65/5.70** | 4.50/5.00 | 2.30/2.10 | 4.80/4.10 | 1.00/1.15 | 2.75/2.95 | Unseen |
| Burmese | **5.50/5.85** | 4.45/5.10 | 5.05/3.70 | 1.80/1.55 | 3.00/3.35 | 1.20/1.45 | Unseen |
| Hindi | **5.55/5.00** | 5.30/4.45 | 3.25/3.70 | 2.65/4.00 | 3.25/2.65 | 1.00/1.20 | Unseen |
| **Summary Statistics** | | | | | | | |
| **Seen Avg** | **5.68/5.37** | 4.06/4.36 | 4.48/3.72 | 2.72/3.30 | 2.27/2.23 | 1.80/2.03 | |
| **Unseen Avg** | **5.57/5.52** | 4.75/4.85 | 3.53/3.17 | 3.08/3.22 | 2.42/2.38 | 1.65/1.87 | |
| **Overall Avg** | **5.65/5.41** | 4.25/4.49 | 4.25/3.56 | 2.82/3.28 | 2.31/2.27 | 1.76/1.99 | |

Table 1: Human Evaluation Results: Comprehensive Weighted Ranking Scores (Higher is Better, **Bold** and underlined values represent the best and second-best results respectively)

### 4.3.1 QUANTITATIVE ANALYSIS

We evaluate MuEx against five SOTA TFS methods: AniTalker Peng et al. (2024), SadTalker Zhang et al. (2023), EchoMimic Chen et al. (2024), AniPortrait Wei et al. (2024), and Hallo2 Cui et al. (2025). All baselines are evaluated on our multilingual dataset using identical preprocessing and evaluation protocols to ensure fair comparison. Table 2 demonstrates MuEx's quantitative superiority across multilingual metrics. Our method achieves state-of-the-art performance on both LSE-D (0.0437) and TMDC (0.756), with a Sync-C score (7.536) that substantially outperforms all competitors and approaches real video quality (8.914). Compared to the strongest baseline Hallo2, MuEx shows remarkable improvements of +3.9% in Sync-C, +2.3% in LSE-D, and +7.2% in TMDC, while traditional methods like SadTalker (6.312) and EchoMimic (6.081) exhibit substantial performance gaps. These results validate our core hypothesis that MuEx architecture with the PG-MoE and PV-Align enables superior multilingual TFS.

| Method | FVD↓ | Sync-C↑ | LSE-D↓ | TMDC↑ |
|---|---|---|---|---|
| Anitalker | 193.762 | 5.594 | 0.0557 | 0.543 |
| SadTalker | 185.382 | 6.312 | 0.0498 | 0.646 |
| EchoMimic | 172.987 | 6.081 | 0.0451 | 0.712 |
| AniPortrait | 236.844 | 4.193 | 0.0492 | 0.654 |
| Hallo2 | **158.625** | 7.469 | 0.0447 | 0.705 |
| MuEx (Ours) | 171.284 | **7.536** | **0.0437** | **0.756** |
| Real video | - | 8.914 | - | - |

Table 2: Comparison of other SOTA methods

| Configuration | FVD↓ | Sync-C↑ | LSE-D↓ | TMDC↑ |
|---|---|---|---|---|
| Baseline (w/o PG-MoE) | 171.547 | 6.921 | 0.0612 | 0.583 |
| + PG-MoE (w/o PV-Align) | 171.392 | 7.089 | 0.0551 | 0.629 |
| + PG-MoE (w/o Guider) | 171.294 | 7.284 | 0.0481 | 0.686 |
| **MuEx (Full)** | **171.284** | **7.536** | **0.0437** | **0.756** |

Table 3: Ablation study on key components

### 4.3.2 QUALITATIVE ANALYSIS

We conducted a comprehensive human evaluation involving 300 participants across 15 language groups. Each group consisted of 20 native speakers who ranked 6 video samples on two dimensions: (1) lip–audio synchronization consistency and (2) teeth visibility clarity and naturalness. Participants ranked methods from 1st to 6th position, with scores calculated using weighted averages where 1st position receives 6 points, 2nd position 5 points, down to 6th position receiving 1 point. Higher scores indicate better performance. The evaluation covered 12 training languages plus 3 unseen languages (Korean, Burmese, Hindi) for zero-shot assessment.

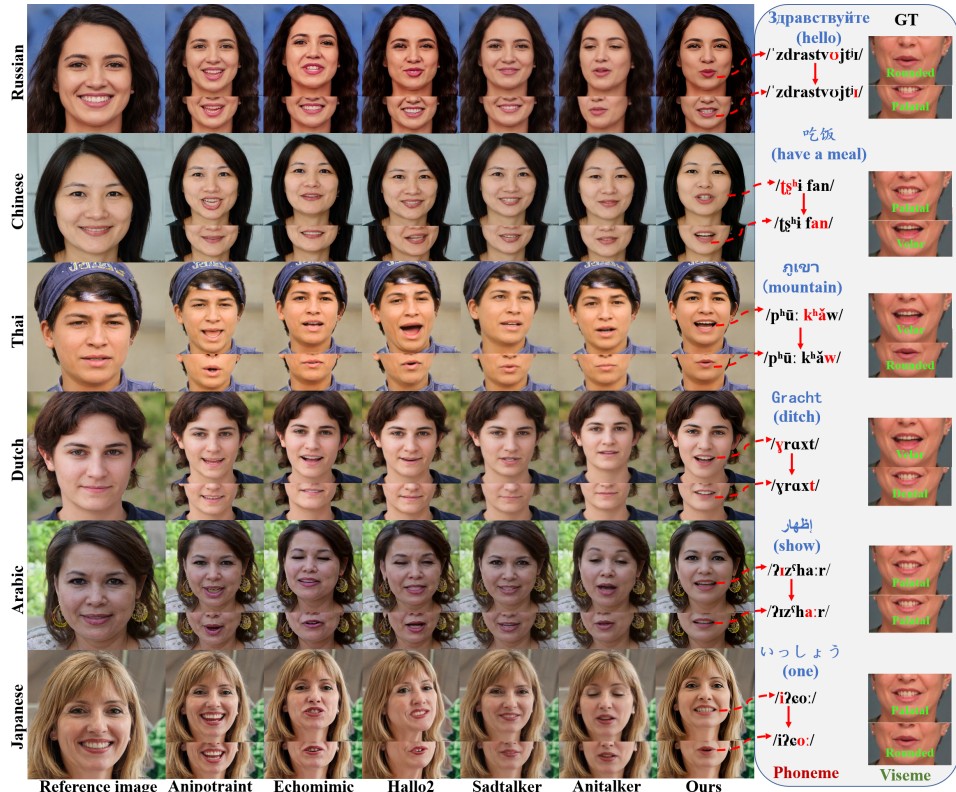

Figure 6: Qualitative comparison. We transcribe test words from different languages into International Phonetic Alphabet (**IPA**) notations and extract specific phoneme video frames to compare with actual mouth shapes (**GT**) during phoneme pronunciation. **Phonemes** represent the basic units of speech sounds, while **visemes** are the corresponding mouth shapes.

As shown in Figures 6 and 7, our method consistently produces more accurate mouth shapes and smoother temporal dynamics across six representative languages, with clear improvements in phoneme-to-viseme alignment compared to baseline methods. The human evaluation results in Table 1 confirm the visual observations, with MUEX scoring highest in both lip–audio synchronization and teeth naturalness across all languages. It shows consistent advantages in tonal languages like Chinese (5.70/5.15), Thai (5.75/5.85), and Vietnamese (5.65/5.30), where other methods struggle. Even for unseen languages, MUEX performs excellently (average scores: 5.57/5.52), demonstrating superior zero-shot generalization.

## 4.4 ABLATION STUDIES

### 4.4.1 CORE COMPONENT ANALYSIS

Table 3 presents a comprehensive ablation study evaluating each key component of our MuEx. The baseline model without PG-MoE shows substantial deficiencies in multilingual synchronization (LSE-D: 0.0612, TMDC: 0.583), confirming that specialized expert routing specifically targets multilingual synchronization challenges. Adding PG-MoE modules without PV-Align provides notable improvements (LSE-D: 0.0551, TMDC: 0.629), demonstrating MoE effectiveness for multilingual variations. The configuration with PG-MoE but without phoneme-guided routing achieves substantial improvements (LSE-D: 0.0481, TMDC: 0.686) through content-based expert selection, yet lacks the language-agnostic inductive bias for optimal cross-linguistic generalization. The complete MuEx delivers optimal performance across all metrics, eliminating explicit language supervision while enabling superior multilingual generalization. Progressive improvements demonstrate framework effectiveness: LSE-D improves by 28.6% and TMDC by 29.7% compared to baseline, validating the superiority of acoustic-articulatory principles for multilingual lip synchronization.

### 4.4.2 HYPERPARAMETER SELECTION

| Hyperparameters | | | Performance Metrics | | | | Efficiency Metrics | |
|---|---|---|---|---|---|---|---|---|
| S | M | K | FVD↓ | Sync-C↑ | LSE-D↓ | TMDC↑ | Memory(GB)↓ | FPS↑ |
| 1 | 2 | 36 | 182.536 | 6.874 | 0.0534 | 0.641 | **1.7** | **31.2** |
| 1 | 3 | 40 | 179.482 | 7.091 | 0.0501 | 0.679 | 2.0 | 28.7 |
| 1 | 4 | 44 | 176.395 | 7.251 | 0.0476 | 0.703 | 2.5 | 25.8 |
| 2 | 3 | 40 | 174.826 | 7.329 | 0.0463 | 0.721 | 2.2 | 26.4 |
| 2 | 4 | 44 | **171.284** | **7.536** | **0.0437** | **0.756** | 2.8 | 22.1 |
| 2 | 5 | 48 | 173.214 | 7.489 | 0.0452 | 0.734 | 3.3 | 19.5 |
| 3 | 4 | 44 | 174.157 | 7.467 | 0.0458 | 0.742 | 3.0 | 20.8 |
| 3 | 6 | 52 | 176.739 | 7.423 | 0.0471 | 0.718 | 4.1 | 16.3 |

Table 4: Trade-off Analysis of Hyperparameter Settings

Table 4 demonstrates a clear performance-efficiency trade-off across different hyperparameter configurations. The optimal configuration ($S = 2, M = 4, K = 44$) achieves the best overall performance (FVD: 171.284, Sync-C: 7.536, LSE-D: 0.0437, TMDC: 0.756) while maintaining reasonable computational efficiency (22.1 FPS, 2.8 GB memory). Lightweight configurations like ($S = 1, M = 2, K = 36$) maximize inference speed (31.2 FPS) and minimize memory usage (1.7 GB) at the cost of significant performance degradation. Notably, performance gains are not monotonic - the ($S = 2, M = 5, K = 48$) configuration underperforms despite higher complexity, while the heaviest configuration ($S = 3, M = 6, K = 52$) shows performance regression, indicating that excessive expert capacity can lead to training instability and suboptimal convergence, emphasizing the importance of balanced hyperparameter selection.

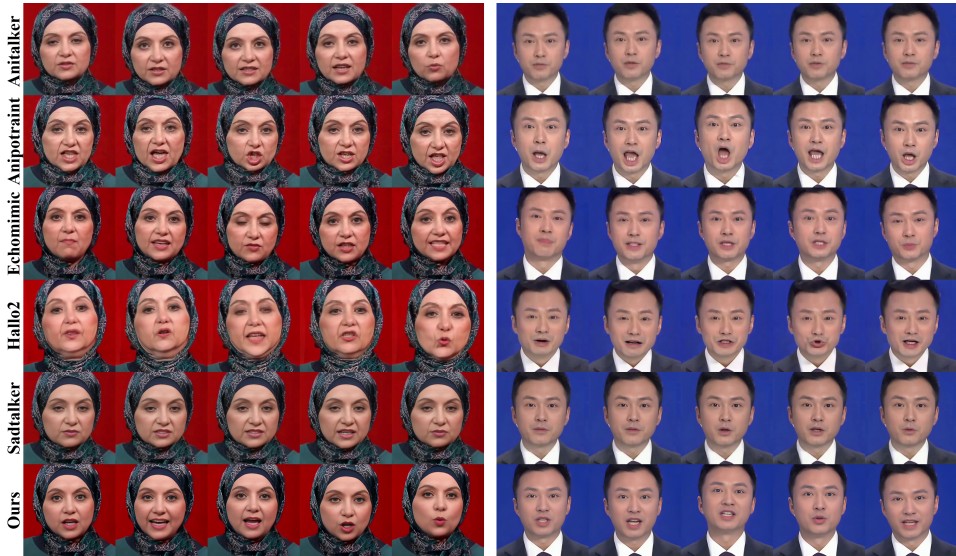

Figure 7: Qualitative comparison between frames generated with Arabic (left) and Chinese (right), highlighting more natural facial expressions and mouth shapes across diverse phonetic contexts

## 5 CONCLUSION

We propose MuEx, a novel framework that addresses the critical limitations of existing TFS methods in multilingual scenarios. Our approach introduces two key innovations: a PV-Align mechanism that creates language-agnostic cross-modal correspondences, and a PG-MoE that enables intelligent expert routing based on audiovisual similarities. Through comprehensive evaluation on our newly constructed MTFB with 12 languages, MuEx demonstrates superior performance in multilingual audiovisual synchronization while maintaining English capabilities and achieving effective zero-shot generalization to unseen languages. This work establishes a new paradigm for multilingual TFS that transcends language boundaries through universal phoneme-viseme principles.

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

# Appendix

## A  LLM USAGE DISCLOSURE

We used a large language model (e.g., ChatGPT) only for language polishing—including grammar correction, wording suggestions, and minor clarity edits. All technical ideas, experiments, analyses, and conclusions are authored and verified by the human authors. The authors carefully reviewed and edited all text; no confidential data were provided to the model, and no content was generated that changes the paper's scientific contributions.

## B  CLUSTERING AND PROTOTYPE DETAILS

### B.1  PROTOTYPE CONSTRUCTION

We construct $K$ prototypes for both phoneme and viseme features, denoted as $\{\mathbf{c}_k^p\}_{k=1}^K$ and $\{\mathbf{c}_k^v\}_{k=1}^K$. In practice, we set $K$ close to the number of phonemes ($K = 44$) and initialize with K-means++ for stable coverage of the feature space. Prototypes are updated periodically (every 10 epochs) to balance adaptability and computational efficiency. Multiple restarts (3 runs) are used and the configuration with the lowest within-cluster distortion is selected.

### B.2  BALANCING STRATEGY

To mitigate the long-tail distribution of phonemes and visemes, we adopt a balancing mechanism. Let $S_{\mathrm{max}}$ denote the maximum cluster capacity and $S_{\mathrm{thr}}$ the effective threshold. For over-represented clusters ($n_k > S_{\mathrm{thr}}$), we keep only the $S_{\mathrm{thr}}$ nearest samples to the centroid. For under-represented clusters, we generate oversampled samples by interpolation:

$$\mathbf{d}_{\mathrm{new}} = \alpha \mathbf{d}_{\mathrm{near}} + (1 - \alpha)\mathbf{c}, \quad \alpha \sim \mathcal{U}(0, 1), \tag{10}$$

where $\mathbf{d}_{\mathrm{near}}$ is the nearest neighbor and $\mathbf{c}$ the centroid.

### B.3  REGULARIZATION

To avoid prototype collapse, we introduce a variance regularization term:

$$\mathcal{L}_{\mathrm{var}} = -\frac{1}{K}\sum_{k=1}^{K}\|\mathbf{c}_k - \bar{\mathbf{c}}\|^2, \tag{11}$$

where $\bar{\mathbf{c}}$ is the mean of all prototypes. This encourages prototypes to spread evenly in the feature space.

## C  DISCRIMINATOR IMPLEMENTATION

### C.1  ARCHITECTURE

We implement the mutual-information (MI) estimator with a lightweight discriminator $D_\psi$ (three-layer MLP). Given a paired input $(\mathbf{x}, \mathbf{y})$ (concatenation or elementwise interaction of features), $D_\psi(\mathbf{x}, \mathbf{y}) \in \mathbb{R}$ outputs a scalar score:

$$D_\psi(\mathbf{x}, \mathbf{y}) = \mathbf{w}_3^\top \sigma(W_2\,\sigma(W_1\,[\mathbf{x}; \mathbf{y}] + \mathbf{b}_1) + \mathbf{b}_2) + b_3, \tag{12}$$

where $W_1 \in \mathbb{R}^{d_h \times (d_x + d_y)}$, $W_2 \in \mathbb{R}^{d_h \times d_h}$, $\mathbf{w}_3 \in \mathbb{R}^{d_h}$, $\sigma(\cdot)$ is ReLU, and $d_h{=}512$ by default. LayerNorm is applied after each hidden layer.

### C.2  INPUTS AND PAIRING

We use two types of inputs depending on the loss term: (i) prototype-level pairs $(\mathbf{q}_t^p, \mathbf{q}_t^v)$ for the positive MI term; (ii) raw feature pairs $(\mathbf{z}_t^p, \mathbf{z}_t^v)$ for the negative MI term. Pairs are formed at matched time steps $t$ after stream synchronization.

### C.3 NEGATIVE SAMPLING

To approximate the product of marginals, we construct mismatched pairs by shuffling:

- **Temporal shuffle**: $(\mathbf{x}_t, \mathbf{y}_{t'})$ with $t' \neq t$ within the same utterance.
- **Utterance shuffle**: across utterances in the same batch.
- **Speaker shuffle**: across speakers when labels are available.

We mix these strategies uniformly unless otherwise specified.

### C.4 JS-BASED MI OBJECTIVE

We adopt the Jensen–Shannon (JS) lower bound with a binary classifier view. Let $\mathcal{P}$ denote matched pairs and $\mathcal{N}$ denote shuffled pairs. The estimator is

$$\widehat{I}^{\mathrm{JS}}(\mathbf{x}, \mathbf{y}) = \mathbb{E}_{(\mathbf{x},\mathbf{y})\sim\mathcal{P}}\left[\log \sigma\big(D_\psi(\mathbf{x},\mathbf{y})\big)\right] + \mathbb{E}_{(\mathbf{x},\mathbf{y})\sim\mathcal{N}}\left[\log\big(1 - \sigma\big(D_\psi(\mathbf{x},\mathbf{y})\big)\big)\right], \tag{13}$$

where $\sigma(\cdot)$ is the logistic sigmoid. This objective is plugged into $\mathcal{L}_{\mathrm{align}} = -\widehat{I}^{\mathrm{JS}}(\mathbf{q}^p, \mathbf{q}^v) + \lambda_{\mathrm{neg}}\widehat{I}^{\mathrm{JS}}(\mathbf{z}^p, \mathbf{z}^v)$.

### C.5 STABILIZATION

To ensure stable training, we employ: (i) gradient clipping with max-norm 5.0; (ii) spectral normalization on $W_1, W_2$; (iii) label smoothing ($\epsilon = 0.05$) for the JS objective; (iv) dropout $p = 0.1$ on hidden layers.

### C.6 TRAINING SCHEDULE

The discriminator and the rest of the model are trained jointly with alternating updates:

1. **MI step**: update $\psi$ by maximizing the JS objective on current features (1 iteration).
2. **Model step**: freeze $\psi$; update encoders, prototypes, and router by minimizing the total loss $\mathcal{L}$ (1 iteration).

We use Adam with learning rate $1\mathrm{e}{-4}$, $\beta_1 = 0.9$, $\beta_2 = 0.999$ for both steps. A linear warm-up of 10k steps is applied, followed by cosine decay.

### C.7 BATCHING AND SAMPLING RATIOS

For each positive pair in a batch, we sample one temporal-shuffle, one utterance-shuffle, and (if available) one speaker-shuffle negative, yielding a 1:3 positive-to-negative ratio. We found this ratio provides a good balance between estimation variance and compute cost.

### C.8 IMPLEMENTATION NOTES

- **Feature normalization**: inputs to $D_\psi$ are LayerNorm-ed to reduce scale sensitivity.
- **Stop-gradient**: prototype assignments used to form $\mathbf{q}^p, \mathbf{q}^v$ feed $D_\psi$ without backpropagating through the argmin; soft assignments can be used if end-to-end gradients are desired.
- **Throughput**: the MI step adds $< 5\%$ wall-clock time per iteration with $d_h = 512$ on A6000-class GPUs.

### C.9 REPRODUCIBILITY

We fix random seeds for shuffling and model initialization, and report results over three runs. Hyperparameters for $D_\psi$ are summarized in Table 5.

| Setting | Value |
|---|---|
| Hidden width $d_h$ | 512 |
| Depth | 3 layers (MLP) |
| Activation | ReLU + LayerNorm |
| Dropout | 0.1 |
| Negatives / positive | 3 |
| Grad clip (max-norm) | 5.0 |
| Optimizer / LR | Adam / 1e−4 |
| Warm-up | 10k steps |
| Decay | Cosine |
| Label smoothing | 0.05 |
| Spectral norm | yes (on $W_1, W_2$) |

Table 5: Discriminator hyperparameters for the JS-based MI estimator.

## D  ROUTING MODULE DETAILS

### D.1  NON-NEGATIVE MAPPING MATRIX

To map pseudo-phoneme s $\mathbf{v}^{(t)}$ to expert-level supervision targets, we introduce a transformation matrix $R \in \mathbb{R}_{\geq 0}^{M \times K}$. To guarantee non-negativity and interpretability, $R$ is parameterized via a softplus activation:

$$R = \text{softplus}(R'), \tag{14}$$

where $R'$ are free parameters. The target distribution for expert supervision is then

$$\mathbf{u}^{(t)} = \text{softmax}\left(\frac{R\mathbf{v}^{(t)}}{\tau_s}\right), \tag{15}$$

where $\tau_s$ is the supervision temperature. This mapping allows each expert to specialize in subsets of articulatory patterns implied by pseudo-phonemes.

### D.2  TEMPERATURE SCHEDULING

We employ two temperatures: $\tau_r$ for routing logits and $\tau_s$ for supervision. Both are linearly annealed from $5.0$ to $0.5$ over the first 50k training steps:

$$\tau \leftarrow \tau_{\text{init}} - (\tau_{\text{init}} - \tau_{\text{final}}) \cdot \frac{\text{step}}{50k}. \tag{16}$$

A higher temperature at early stages encourages exploration of multiple experts, while a lower temperature later sharpens the distribution for stable specialization.

### D.3  EXPERT USAGE ANALYSIS

To avoid expert collapse, we monitor the usage counts $n_i$ for each expert $i$. Recall the utilization regularization in the main paper:

$$\mathcal{L}_{\text{util}} = \frac{M}{B} \sum_{i=1}^{M} \left(\frac{n_i}{B}\right)^2,$$

where $B$ is the batch size. This term encourages balanced load across experts. Additionally, the entropy regularization

$$\mathcal{L}_{\text{ent}} = -\sum_{i \in \mathcal{Q}} w_i \log w_i$$

prevents over-confident assignments among the selected top-$S$ experts.

## E  TRAINING HYPERPARAMETERS AND SETUP

This section documents the configuration needed to reproduce our results, including optimization details, data preprocessing, schedules, and hardware.

### E.1 Optimization

We train all models end-to-end with Adam:

- Optimizer: Adam ($\beta_1{=}0.9,\ \beta_2{=}0.999$), $\epsilon{=}1\mathrm{e}{-}8$.
- Base learning rate: $1\mathrm{e}{-}4$ for encoders, prototypes, router, discriminator, and generator.
- Weight decay: $1\mathrm{e}{-}4$ on all modules except normalization layers and bias terms.
- Gradient clipping: global norm $5.0$.
- Mixed precision: FP16 for all modules (AMP); loss-scaling is dynamic.

### E.2 Learning-Rate and Temperature Schedules

We use a warm-up followed by cosine decay and apply temperature annealing for routing/supervision:

- LR schedule: linear warm-up for the first 10k steps, then cosine decay to $10\%$ of the base LR by the final step.
- Routing temperature $\tau_r$: linearly annealed $5.0 \rightarrow 0.5$ over the first 50k steps.
- Supervision temperature $\tau_s$: linearly annealed $5.0 \rightarrow 0.5$ over the first 50k steps.
- Discriminator LR follows the same schedule as the main model.

### E.3 Loss Weights

Unless otherwise noted, we use the following default weights:

$$\lambda_1 = 1.0, \quad \lambda_p = 0.1, \quad \lambda_t = 0.1, \quad \lambda_{\mathrm{neg}} = 0.1, \quad \lambda_{\mathrm{util}} = 0.05, \quad \lambda_{\mathrm{ent}} = 0.05, \quad \lambda_{\mathrm{task}} = 1.0.$$

These values balance pixel fidelity, perceptual realism, temporal stability, alignment strength, and routing regularization.

### E.4 Batching, Sequence Length, and Resolution

- Batch size $B{=}64$ sequences per step (accumulated if memory-limited).
- Sequence length $T{=}200$ frames; we use random contiguous crops for training.
- Frame resolution $512{\times}512$; audio is processed at $16\,\mathrm{kHz}$ with standard mel front-ends.
- Gradient accumulation: $N_{\mathrm{acc}}{=}1{\sim}4$ depending on GPU memory.

### E.5 Data Preprocessing and Augmentation

- Video: face detection and alignment (5-point landmarks), center-crop and resize to $512{\times}512$.
- Audio: pre-emphasis, STFT, 80-dim mel-spectrograms; mean-variance normalization per utterance.
- Augmentations: random horizontal flip (p=0.5), random brightness/contrast jitter (small range), time masking on mel features (SpecAugment-style) for robustness.
- Cross-lingual setup: language labels are *not* used by the model; splits are language-disjoint for evaluation.

### E.6 Hardware and Runtime

- Hardware: $4{\times}$A6000 (48 GB) GPUs.
- Training length: 60k–80k steps (depends on dataset size).
- Checkpointing: save every 2k steps; keep top-3 by validation LSE-C.