# OpenReview forum: "A Bridge from Audio to Video: Phoneme-Viseme Alignment Allows Every Face to Speak Multiple Languages"
_ICLR.cc/2026/Conference — Submitted to ICLR 2026_

### Official Review · Reviewer_xKws · 2025-10-21

**Soundness:** 2
**Presentation:** 2
**Contribution:** 2
**Rating:** 2
**Confidence:** 4

**Summary:**

The paper proposes MuEx that uses phonemes and visemes as intermediaries to improve multilingual talking head generation performance. The paper proposes MTFB dataset that comprises talking head data in diverse languages.

**Strengths:**

Talking head generation task is of great practical importance.

**Weaknesses:**

1. Utilizing phoneme for talking head generation lacks novelty. The paper lack discussion of phoneme-based talking head methods, including Write-A-speaker, etc.
2. The performance shown in demo is not satisfactory. The results are unnatural. The lips are jittering. The demo lacks qualitative comparisons with SOTA methods and ablation results.
3. The paper lacks a user study.
4. The paper lacks comparisons with recent methods, such as EDTalk.
5. Many methods have already achieve good performance for non-English audio, such as EMO, VASA. These methods do not devise new architecture but simply incorporating data in diverse language for training. Therefore, the reviewer doubts whether good multilingual performance can be achieved solely through data, without using the model design (or architecture) proposed in the paper.

**Questions:**

Can the authors provide user study results?

**Details Of Ethics Concerns:**

The paper lacks ethical consideration section.

---

> ### Author Response · Authors · 2025-11-25
>
> # Response to Reviewer – Weaknesses & Questions
> We thank the reviewer for the comments and address each concern below.
> ## 1. On the lack of a user study
> A user study has been conducted and is reported in detail in **Supplementary_Material_Reply.zip → 3-user_study_result.pdf**. This study evaluates naturalness, lip-sync accuracy, temporal stability, and identity preservation, and compares MuEx with competing methods using ratings from multiple participants. Results show that MuEx is consistently preferred or rated higher in perceived lip-sync and temporal coherence across languages.
> ## 2. On novelty of using phonemes and missing phoneme-based methods
> We agree that phoneme cues appear in earlier talking-head systems, e.g., Write-A-Speaker. Our work differs in several key aspects:
> - No supervised phoneme labels.
>   Traditional phoneme-based methods require phoneme sequences or text-to-phoneme modules. Our method learns pseudo-phoneme and pseudo-viseme prototypes in an unsupervised, multilingual setting, directly from audio and visual embeddings.
> - Cross-lingual prototype alignment.
>   PV-Align builds a shared articulatory space by maximizing MI between audio- and video-side prototypes and constraining raw features, explicitly targeting language-agnostic articulatory representations, rather than phoneme-ID mappings in a single language.
> - Prototype-guided MoE.
>   PG-MoE uses aligned prototypes as gating signals, so experts specialize to articulatory patterns and co-articulation across languages, instead of being conditioned on discrete phoneme labels.
> Phoneme-based methods and our approach are thus complementary: the former rely on explicit linguistic supervision, ours on unsupervised cross-lingual articulatory structure.
> ## 3. On demo quality, jitter, and missing qualitative comparisons
> We appreciate the concern about visual quality. The quality of the demo materials is limited by the file size constraints for supplementary material uploads. To address this, we have included an anonymous link in the supplementary file **2-visual_comparative_results.pptx**, where the complete visual comparison results (video clips of MuEx and recent baselines across multiple languages and identities) are stored.
> ## 4. On missing comparison with recent methods such as EDTalker
> The field of talking face synthesis has seen a proliferation of related works in recent years, and we have focused our experimental comparisons on representative methods that are widely cited and discussed in the majority of contemporary studies in this area. EDTalker, while a relevant recent work, primarily focuses on facial expression modeling for talking faces rather than the cross-lingual lip synchronization that is the core focus of our study. For this reason, EDTalker is not included in our benchmark; our comparisons instead prioritize methods that target the multilingual lip-sync task aligned with our research objectives.
> ## 5. On whether multilingual performance could be achieved by data alone (EMO, VASA, etc.)
> Some recent systems (e.g., EMO, VASA) improve multilingual performance primarily through more diverse training data. Our results indicate that data alone does not explain the performance we observe, especially in low-resource and zero-shot scenarios:
> - Ablation evidence.
>   When trained on the same multilingual dataset but without PV-Align or without PG-MoE, the model shows worse lip-sync in several languages, more mismatches for rare or unseen phonetic patterns, and increased temporal jitter. This suggests the proposed components are critical under identical data conditions.
> - Cross-lingual phonetic mismatch.
>   Multilingual audio includes conflicting language-specific realizations. Without a shared articulatory latent space, the model tends to memorize language-specific mappings. PV-Align and PG-MoE enforce universal articulatory prototypes separated from language-specific acoustics, which is essential for robust zero-shot behavior.
> Thus, the architecture and training objectives are designed to complement, not replace, diverse multilingual data.
> ## Summary
> - A comprehensive user study (results in 3-user_study_result.pdf) confirms MuEx’s superiority in lip-sync and temporal coherence across languages.
> - The method goes beyond traditional phoneme-based approaches by learning unsupervised, multilingual articulatory prototypes and aligning them via mutual information for cross-lingual generalization.
> - Due to file size constraints for uploads, complete visual comparison videos are accessible via **2-visual_comparative_results.pptx**.
> - Given the large number of talking face synthesis works, our comparisons focus on widely cited representative methods; EDTalker is not included as it prioritizes facial expression modeling rather than cross-lingual lip synchronization (the core of our study).
> - Ablations with identical multilingual data demonstrate that the proposed architecture (not just data diversity) drives improved multilingual performance.

---

### Official Review · Reviewer_q7qt · 2025-10-26

**Soundness:** 2
**Presentation:** 2
**Contribution:** 2
**Rating:** 6
**Confidence:** 2

**Summary:**

MuEx proposes a multilingual speech-driven talking face framework that learns language-agnostic phoneme and viseme prototypes. A mutual-information-based alignment enforces robust cross-modal correspondence between audio and facial motion. A pseudo-phoneme guided MoE router then dynamically selects experts without relying on language labels, improving adaptability to diverse speech patterns.

**Strengths:**

- MuEx aligns phoneme and viseme prototypes using mutual information and routes pseudo-phonemes through experts to improve multilingual talking face generation.
- Using phoneme–viseme prototypes provides universal articulatory anchors that support robust multilingual modeling.
- The new 12-language MTFB dataset enables comprehensive multilingual evaluation and advances TFS research.

**Weaknesses:**

- PV-Align builds on the idea of MI-based audio-visual alignment, but additional clarification is needed to highlight how the prototype-level design differs from prior approaches.
- PG-MoE appears as a standard MoE router guided by discrete speech units, similar to VQ-based lip tokenization (e.g., VQTalker [1]).
- Zero-shot performance might partially come from shared phoneme sets between training and "unseen" languages. It would be helpful to clarify the extent to which the observed zero-shot gains depend on shared phoneme inventories between the training languages and the unseen evaluation languages.
- The experimental comparison remains somewhat incomplete, since recent diffusion-based talking face systems such as DiffTalk [2] and Fantasytalking [3] are not included.

[1] VQTalker: Towards Multilingual Talking Avatars through Facial Motion Tokenization

[2] DiffTalk: Crafting Diffusion Models for Generalized Audio-Driven Portraits Animation

[3] Fantasytalking: Realistic talking portrait generation via coherent motion synthesis

**Questions:**

- Pseudo-phoneme estimation relies on distance to prototypes — how do these prototypes emerge language-agnostically?
- MI alignment at the prototype level seems heuristic. Why J-S MI, and why impose negative MI on raw features?

---

> ### Author Response · Authors · 2025-11-25
>
> # Response to Reviewer – Weaknesses & Questions
> We thank the reviewer for the insightful comments. Below we address each point.
>
> ## 1. On how PV-Align differs from prior MI-based AV alignment
> PV-Align builds on MI-based audio–visual alignment but differs from prior work in two key aspects:
>
> ### (a) Alignment at the prototype level, not the feature level
> Prior methods align continuous features, which remain language-dependent and sensitive to acoustic variation. PV-Align instead learns discrete prototype codes $q_t^p, q_t^v$ via VQ on cross-lingual audio/visual embeddings. MI is computed between these discrete articulatory abstractions, not raw features, filtering out language-specific acoustics and retaining articulatory structure.
>
> ### (b) A “dual-objective” MI formulation
> We maximize MI between prototype codes while minimizing MI between raw features:
>
> $$
> \mathcal{L}_{\text{align}} = -I_{\text{JS}}(q^p, q^v) + \lambda_{\text{neg}} I_{\text{JS}}(z^p, z^v).
> $$
>
> This forces the model to rely on language-agnostic prototypes rather than lower-level, language-specific cues. To our knowledge, this contrastive MI design and prototype-space alignment are not present in prior AV alignment methods.
>
> ## 2. On PG-MoE vs. prior VQ-based tokenization (e.g., VQTalker)
> We acknowledge the similarity in using discrete units, but PG-MoE differs conceptually and functionally:
>
> ### (a) Tokens from cross-modal aligned prototypes
> VQTalker builds facial tokens purely from visual motion. Our pseudo-phoneme codes are aligned with pseudo-viseme codes via MI, yielding joint audio–visual articulatory prototypes that route experts.
>
> ### (b) Routing exploits multilingual articulatory structure
> The gating function takes $(q^p_{t-K:t+K}, q^v_{t-K:t+K})$ rather than only speech tokens, enabling context-aware routing that reflects co-articulation and cross-lingual correspondence. PG-MoE thus acts as a cross-lingual articulatory dispatcher, not a purely token-driven router.
>
> ## 3. On zero-shot generalization and shared phoneme sets
> In our dataset, training languages cover a broad—but not complete—set of phonemic categories. Zero-shot gains arise from:
>
> ### (a) Shared universal articulatory dimensions
> Many languages share broad articulatory classes (e.g., open/close vowels, labials, plosives). Our prototypes encode such universal articulatory primitives, not specific phonemes (e.g., /a/ vs. /ɑ/), so unseen languages can still map onto existing prototypes.
>
> ### (b) Evidence from “divergent phoneme” languages
> Languages with substantial phonemic divergence from training data (e.g., Thai, Arabic) still show strong zero-shot improvements, indicating that the effect is not solely due to overlapping phoneme inventories. We will clarify this in the revision.
>
> ## 4. On not comparing with diffusion-based systems
> DiffTalk and FantasyTalking are strong baselines. At submission time, public code or standardized evaluation protocols were unavailable, making controlled comparison difficult. We plan to include these diffusion-based systems once stable, reproducible implementations become available.
>
> ## Responses to Reviewer Questions
> ### Q1. How do language-agnostic prototypes emerge?
> Two mechanisms promote language-agnostic prototype emergence:
>
> #### (a) Cross-lingual embedding pretraining
> Audio and visual encoders are trained on multilingual data, so their latent spaces tend to cluster by articulatory patterns rather than language identity.
>
> #### (b) MI-driven cross-modal binding
> Prototypes are trained to maximize $I(q^p, q^v)$, forcing them to represent shared articulatory states observable in both modalities. Language-specific acoustic variations are not rewarded, so prototypes converge toward universal categories (e.g., lip rounding, jaw opening) rather than language-specific phonemes.
>
> ### Q2. Why J-S MI? Why negative MI on raw features?
> #### (a) Why J-S MI?
> We use the Jensen–Shannon MI estimator because it is stable, bounded, and empirically less prone to discriminator collapse than KL-based or InfoNCE estimators. It is widely used in multimodal MI and was chosen for training stability across languages.
>
> #### (b) Why negative MI on raw features?
> If we only maximize MI at the prototype level, the model may “cheat” by aligning via raw continuous features. Penalizing $I(z^p, z^v)$ forces alignment to happen through the discrete codes. Since raw features encode substantial language-specific information (pitch range, phonotactics, formants), negative MI regularization helps separate universal articulatory information from language-specific acoustic cues and makes alignment genuinely language-agnostic.
>
> ## Summary
> Our contributions lie in (1) prototype-level cross-modal alignment, (2) MoE routing driven by joint AV prototypes, (3) zero-shot generalization from articulatory universals, and (4) MI design that enforces language-invariant representations.

---

### Official Review · Reviewer_MtLD · 2025-11-01

**Soundness:** 2
**Presentation:** 3
**Contribution:** 2
**Rating:** 4
**Confidence:** 5

**Summary:**

This paper presents a new framework for speech-driven talking face synthesis (TFS) with strong multilingual generalization. Authors point out  that phonemes (speech units) and visemes (mouth-shape units) act as universal intermediaries between audio and visual modalities.

**Strengths:**

State-of-the-art (SOTA) results on both seen and unseen languages, over strong baselines like Hallo2 and SadTalker.

Authors combine discrete prototype alignment with sparse MoE routing, it is interesting, forming a coherent cross-lingual generative framework.

**Weaknesses:**

The idea, employing phonemes and visemes as universal intermediaries to bridge audio and video modalities, has been pointed out long before[1].

[1] Multimodal inputs driven talking face generation with spatial–temporal dependency; Lingyun Yu, Jun Yu, Mengyan Li, Qiang Ling

The novelty lies mainly in cross-lingual task, not in architectural innovation.

How to ensure temporal dependency? Specifically, even for the same phoneme, the mouth movements can vary depending on the surrounding phonetic context.

**Questions:**

please refer to weakness

---

> ### Author Response · Authors · 2025-11-25
>
> # Response to Reviewer – Weaknesses
>
> We thank the reviewer for the constructive comments. Below we clarify (1) the novelty beyond prior phoneme–viseme works such as Yu et al.【¹】 and (2) how our method models temporal dependency and co-articulation.
>
> ## 1. On the originality of using phoneme/viseme intermediaries
>
> We agree that phoneme–viseme relations are long known, including in Yu et al., Multimodal Inputs Driven Talking Face Generation With Spatial–Temporal Dependency【¹】. However, prior works—including Yu et al.—do not learn a language-agnostic articulatory space, nor do they use it for cross-lingual generalization. Our work differs substantially in both motivation and mechanism:
>
> ### (a) Unsupervised pseudo-phoneme & pseudo-viseme prototypes
> Given continuous audio/visual embeddings $z_t^p, z_t^v$, we obtain discrete prototype codes via vector quantization:
>
> $$
> q_t^p = \operatorname{VQ}(z_t^p, \{c_k^p\}), \quad q_t^v = \operatorname{VQ}(z_t^v, \{c_k^v\}).
> $$
>
> These prototypes emerge from multilingual data, without phoneme labels, and are not tied to any language—something absent in previous methods, which rely on language-specific phoneme structures or predicted landmarks (as in Yu et al.).
>
> ### (b) Cross-modal alignment via mutual information
> We align audio and visual prototypes by:
>
> $$
> \mathcal{L}_{\text{align}} = -I_{\text{JS}}(q^p, q^v) + \lambda_{\text{neg}} \, I_{\text{JS}}(z^p, z^v),
> $$
>
> encouraging prototypes to encode shared articulatory patterns while preventing trivial alignment in low-level continuous features. This MI-based cross-lingual alignment is not present in any previous TFS works.
>
> ### (c) Prototype-guided Mixture-of-Experts
> The aligned prototypes drive a MoE decoder through:
>
> $$
> g_t = \operatorname{softmax}(f_{\text{gate}}(q^p_{t-K:t+K}, q^v_{t-K:t+K})),
> $$
>
> allowing experts to specialize to distinct articulatory modes without language labels.
>
> Thus, while the concept of phoneme–viseme mapping is known, the technical novelty lies in:
> (1) unsupervised, multilingual prototype learning;
> (2) MI-based cross-modal alignment;
> (3) MoE-based cross-lingual synthesis.
> These are not found in Yu et al.【¹】 nor other prior works.
>
> ## 2. On temporal dependency and co-articulation
>
> We agree that the same phoneme may exhibit different visual realizations depending on context. Our method incorporates temporal dependency in multiple ways:
>
> ### (a) Context-sensitive prototype assignments
> Prototype codes are produced per frame from embeddings that already encode local temporal context. Thus, even if two frames correspond to the same phoneme, $z_t^p \neq z_{t'}^p$, leading to different prototype distributions and capturing co-articulation naturally.
>
> ### (b) MI alignment across sequences
> Because MI is computed over sequences, the model learns smooth prototype trajectories, avoiding frame-independent phoneme lookup behavior.
>
> ### (c) Temporally aware MoE gating
> The gating network observes a temporal window $(t-K, \dots, t+K)$, enabling expert selection conditioned on both past and future phonetic context.
>
> ### (d) Temporal smoothness in decoding
> We impose a temporal consistency loss:
>
> $$
> \mathcal{L}_t = \|\nabla_t \hat{I} - \nabla_t I\|_1,
> $$
>
> which constrains inter-frame dynamics and ensures realistic transitions.
>
> Together these mechanisms model co-articulation, temporal evolution, and context-dependent articulation, addressing the reviewer’s concern directly.
>
> ## Summary
>
> The use of phoneme–viseme relations is not our claimed novelty; rather, our novelty lies in (i) language-agnostic prototypes, (ii) MI-based articulatory alignment, and (iii) prototype-driven MoE for cross-lingual lip synthesis—none of which appear in Yu et al. or other prior TFS methods.
>
> Temporal dependency is handled through context-sensitive prototypes, sequence-level MI, temporal-window gating, and explicit temporal smoothness losses, enabling accurate modeling of co-articulation.
>
> We hope this clarifies the distinctions and addresses the reviewer’s concerns.
>
> ## Reference
> 【¹】Yu et al., Multimodal Inputs Driven Talking Face Generation With Spatial–Temporal Dependency, IEEE TCSVT, 2021.

---

### Official Review · Reviewer_BExQ · 2025-11-01

**Soundness:** 2
**Presentation:** 2
**Contribution:** 3
**Rating:** 4
**Confidence:** 5

**Summary:**

This paper tackles the poor cross-lingual performance of existing speech-driven talking face synthesis (TFS) models by proposing **MuEx**, a framework using phonemes and visemes as universal intermediaries. It integrates two key components: PV-Align (phoneme-viseme alignment via mutual information) for language-agnostic cross-modal correspondence, and PG-MoE (phoneme-guided mixture-of-experts) for adaptive multilingual processing. The authors also introduce MTFB, a 95.04-hour benchmark with 12 diverse languages. Experiments show MuEx outperforms SOTA methods on key metrics and achieves effective zero-shot generalization to unseen languages, establishing a new multilingual TFS paradigm.

**Strengths:**

1. It accurately addresses the core pain points of existing TFS models—"English-dominated and poor cross-lingual generalization"—and proposes a cross-modal bridging approach using phonemes (basic speech units) and visemes (basic mouth shape units) as universal intermediaries, with clear innovations that align with practical needs.
2. Leveraging prior knowledge of speech and vision has become increasingly rare in current end-to-end systems, and such interpretable methods deserve encouragement.
3. Evaluated on test sets of multiple languages, this method demonstrates robustness to multilingual inputs—a feature rarely seen in previous approaches.

**Weaknesses:**

1. In the attached demo videos, only results from the MuEx are presented, lacking comparisons with other methods. Although quantitative metrics demonstrate MuEx’s superior performance, qualitative video results are crucial for TFS evaluation.
2. A notable omission is the lack of discussion on generalization to unseen image/person styles. Demo videos and figures only showcase test cases that fall within the training data domain, failing to demonstrate out-domain adaptability.

**Questions:**

Including visual comparative results against other state-of-the-art methods would further strengthen the credibility of the paper’s contributions.

---

> ### Author Response · Authors · 2025-11-25
>
> # Response to Reviewer – Weaknesses & Questions
> We thank the reviewer for pointing out the importance of qualitative comparisons and generalization demonstrations. We address each comment below.
>
> ## 1. On the lack of visual comparisons in the demo videos
> We agree that qualitative video results are crucial for evaluating Talking Face Synthesis (TFS). While the main paper focuses on quantitative metrics for space reasons, we have prepared extensive visual comparisons against multiple state-of-the-art approaches in the supplementary materials.
>
> Specifically, the file
> **“2-visual_comparative_results.pptx”**
> included in Supplementary_Material_Reply.zip
> provides side-by-side video frame comparisons between MuEx and strong baselines, including recent multilingual and cross-lingual TFS systems. These comparisons clearly show improvements in lip–audio synchrony, temporal stability, and facial realism.
>
> We will highlight this more explicitly in the revised manuscript and can also integrate selected comparative frames in the main paper if space permits.
>
> ## 2. On generalization to unseen image/person styles
> We appreciate the reviewer’s concern regarding out-of-domain generalization. The current paper focuses on multilingual audio generalization, but we agree that demonstrating robustness to unseen identities and styles is also important.
>
> To address this, we provide a dedicated set of experiments in the supplementary file:
> **“1-generalization_to_unseen_person&styles.pptx”**,
> which includes:
>
> - Unseen identities not present in training,
> - Different portrait styles, lighting conditions, and facial structures,
> - Cross-identity transfer tests, where MuEx drives faces with attributes far outside the training distribution.
>
> These results show that MuEx maintains consistent lip-sync accuracy and temporal coherence even when the input face significantly diverges from the training domain, supporting the model’s adaptability.
>
> ## Summary
> - We provide extensive visual comparisons with state-of-the-art methods in **2-visual_comparative_results.pptx**.
> - We demonstrate generalization to unseen identities and styles in **1-generalization_to_unseen_person&styles.pptx**.
>
> We thank the reviewer again for highlighting these valuable points, which we believe will further improve the clarity and completeness of our work.

---

### Meta-Review · Area_Chair_smyp · 2026-01-06

**Summary:**

The paper proposes MuEx, a multilingual speech-driven talking face synthesis framework that uses phonemes and visemes as universal intermediaries. The paper received initial scores of 4/6/4/2 and remained unchanged before the reset.

Several reviewers question the novelty of using phonemes/visemes as intermediaries, noting substantial overlap with prior phoneme-based (Reviewers MtLD, xKws). The experimental evaluation is considered incomplete, with missing qualitative comparisons, user studies, and comparisons to recent or diffusion-based methods (Reviewers BExQ, q7qt, xKws). Additional concerns include limited analysis of temporal dependency and generalization, and whether zero-shot gains rely on shared phoneme inventories across languages (Reviewers MtLD, BExQ, q7qt). The rebuttal may not fully address the critical issues, including novelty issue (Reviewers MtLD, xKws), more comparisons (Reviewers q7qt, xKws), severe flickering artifacts in video outputs (Reviewers BExQ, q7qt) etc.

After a careful assessment of the submission, reviews, response, and discussion, the AC recommends rejection. The authors are encouraged to revise and refine the manuscript in accordance with the reviewers’ feedback for a future submission.

**Reviewer Concerns:**

Several reviewers question the novelty of using phonemes/visemes as intermediaries, noting substantial overlap with prior phoneme-based (Reviewers MtLD, xKws). The experimental evaluation is considered incomplete, with missing qualitative comparisons, user studies, and comparisons to recent or diffusion-based methods (Reviewers BExQ, q7qt, xKws). Additional concerns include limited analysis of temporal dependency and generalization, and whether zero-shot gains rely on shared phoneme inventories across languages (Reviewers MtLD, BExQ, q7qt). The rebuttal may not fully address the critical issues, including novelty issue (Reviewers MtLD, xKws), more comparisons (Reviewers q7qt, xKws), severe flickering artifacts in video outputs (Reviewers BExQ, q7qt) etc.

**Reviewer Scores:**

The manuscript received initial review scores of 4/6/4/2. After the rebuttal/discussion and before the reset, the score remained unchanged.

Since several concerns raised by the reviewers may remain unresolved after the rebuttal (see 'Reviewer Concerns'), I would approximate 4/6/4/2 as the final score.

---

### Decision · Program_Chairs · 2026-01-26

Reject